# MEMS Deformable Mirrors for Space-Based High-Contrast Imaging

**DOI:** 10.3390/mi10060366

**Published:** 2019-05-31

**Authors:** Rachel E. Morgan, Ewan S. Douglas, Gregory W. Allan, Paul Bierden, Supriya Chakrabarti, Timothy Cook, Mark Egan, Gabor Furesz, Jennifer N. Gubner, Tyler D. Groff, Christian A. Haughwout, Bobby G. Holden, Christopher B. Mendillo, Mireille Ouellet, Paula do Vale Pereira, Abigail J. Stein, Simon Thibault, Xingtao Wu, Yeyuan Xin, Kerri L. Cahoy

**Affiliations:** 1Department of Aeronautics and Astronautics, Massachusetts Institute of Technology, Cambridge, MA 02139, USA; gregallan@mit.edu (G.W.A.); jgubner@wellesley.edu (J.N.G.); chaughwo@mit.edu (C.A.H.); bgholden@mit.edu (B.G.H.); paulavp@mit.edu (P.d.V.P.); ajmstein@mit.edu (A.J.S.); yinzi@mit.edu (Y.X.); 2Department of Astronomy, Steward Observatory, University of Arizona, Tucson, AZ 85719, USA; 3Boston Micromachines Corporation, Cambridge, MA 02138, USA; pab@bostonmicromachines.com; 4Department of Physics, University of Massachusetts Lowell, Lowell, MA 01854, USA; Supriya_Chakrabarti@uml.edu (S.C.); Timothy_Cook@uml.edu (T.C.); 5Lowell Center for Space Science and Technology, University of Lowell, Lowell, MA 01854, USA; christopher_mendillo@uml.edu; 6Department of Physics, Massachusetts Institute of Technology, Cambridge, MA 02139, USA; mdegan@mit.edu (M.E.); gfuresz@mit.edu (G.F.); 7Department of Physics, Wellesley College, Wellesley, MA 02481, USA; 8NASA Goddard Space Flight Center, Greenbelt, MD 20771, USA; Tyler.d.groff@nasa.gov; 9Universite Laval, Québec City, QC G1V 0A6, Canada; mireille.ouellet.4@ulaval.ca (M.O.); simon.thibault@phy.ulaval.ca (S.T.); 10Microscale Inc., Woburn, MA 01801, USA; xwu@microscaleinc.com; 11Department of Earth, Atmospheric, and Planetary Sciences, Massachusetts Institute of Technology, Cambridge, MA 02139, USA

**Keywords:** MEMS deformable mirrors, exoplanet direct imaging, wavefront control, space telescope technology

## Abstract

Micro-Electro-Mechanical Systems (MEMS) Deformable Mirrors (DMs) enable precise wavefront control for optical systems. This technology can be used to meet the extreme wavefront control requirements for high contrast imaging of exoplanets with coronagraph instruments. MEMS DM technology is being demonstrated and developed in preparation for future exoplanet high contrast imaging space telescopes, including the Wide Field Infrared Survey Telescope (WFIRST) mission which supported the development of a 2040 actuator MEMS DM. In this paper, we discuss ground testing results and several projects which demonstrate the operation of MEMS DMs in the space environment. The missions include the Planet Imaging Concept Testbed Using a Recoverable Experiment (PICTURE) sounding rocket (launched 2011), the Planet Imaging Coronagraphic Technology Using a Reconfigurable Experimental Base (PICTURE-B) sounding rocket (launched 2015), the Planetary Imaging Concept Testbed Using a Recoverable Experiment - Coronagraph (PICTURE-C) high altitude balloon (expected launch 2019), the High Contrast Imaging Balloon System (HiCIBaS) high altitude balloon (launched 2018), and the Deformable Mirror Demonstration Mission (DeMi) CubeSat mission (expected launch late 2019). We summarize results from the previously flown missions and objectives for the missions that are next on the pad. PICTURE had technical difficulties with the sounding rocket telemetry system. PICTURE-B demonstrated functionality at >100 km altitude after the payload experienced 12-g RMS (Vehicle Level 2) test and sounding rocket launch loads. The PICTURE-C balloon aims to demonstrate $10^{-7}$ contrast using a vector vortex coronagraph, image plane wavefront sensor, and a 952 actuator MEMS DM. The HiClBaS flight experienced a DM cabling issue, but the 37-segment hexagonal piston-tip-tilt DM is operational post-flight. The DeMi mission aims to demonstrate wavefront control to a precision of less than 100 nm RMS in space with a 140 actuator MEMS DM.

## 1. Introduction

Micro-Electro-Mechanical Systems (MEMS) Deformable Mirrors (DMs) are a promising technology enabling precise wavefront control for high-contrast imaging systems. MEMS DMs offer high-actuator density, large stroke, and precise control in a small, low-power form factor, which makes them suitable for space-based wavefront control applications, such as space telescope coronagraph instruments for exoplanet direct imaging [1,2]. For instance, the Boston Micromachines Corporation (BMC) 4K DM has 4092 actuators across a 25 mm × 25 mm aperture with a stroke of 4 μm [3] and is used on the ground for extreme adaptive optics on the Gemini Planet Imager [4].

In this paper, we summarize recent and planned astrophysics missions that will operate MEMS DMs at high altitudes and in the space environment. We briefly motivate the use of MEMS DMs for high contrast imaging of exoplanets and describe the MEMS DM design and manufacturing process at a high level. We then discuss design approaches and considerations for adaptive optics (AO) systems for space-based observatories. We introduce ground testbed laboratory results as well as sounding rocket, high-altitude balloon, and CubeSat flight missions that incorporate MEMS DMs into their payloads, and describe mission performance goals and results to date. We close with a discussion of next steps to incorporate MEMS DMs into larger space platforms with longer mission lifetimes and more challenging performance requirements.

## 2. Background

### 2.1. High Contrast Imaging of Exoplanets

High-contrast imaging can enable detailed characterization of exoplanets by gathering precise astrometric data [5] and measuring a planet’s atmosphere through spectroscopy [6]. This data will be key for constraining the orbits of exoplanets [7] and detecting biosignatures in order to assess the potential habitability of exoplanets [8]. A detailed overview of the techniques for exoplanet direct imaging can be found in Traub and Oppenheimer’s Direct Imaging chapter of Seager’s book Exoplanets [9]. High contrast imaging can also detect and characterize circumstellar debris disk structure to understand solar system formation [10,11].

Directly imaging Earth-like exoplanets is an immense technical challenge. Resolving an Earth-like planet near a Sun-like star with a telescope requires instruments that are able to image at extremely high contrasts of $10^{-10}$ [12]. Achieving this contrast level with a coronagraph instrument, which blocks out the light of the target star so the dim companion planet is visible, requires an adaptive optics system capable of picometer-level wavefront control [13]. Deformable mirrors can be used in coronagraph systems to control the wavefront in order to prevent speckles and stray light from degrading the contrast [13,14,15]. Picometer-level control has been demonstrated at the NASA Jet Propulsion Laboratory (JPL) High Contrast Imaging Testbed [16]. This level of stability refers to the average surface flatness over many spatial modes, not the flatness of individual atoms [9].

### 2.2. Deformable Mirror Technology

An adaptive optics system corrects wavefront errors in a telescope system in order to improve image quality and contrast. It was first proposed in 1953 to correct for errors due to atmospheric turbulence for ground-based observatories [17]. For space telescopes, adaptive optics is proposed to correct wavefront errors due to thermal and mechanical effects in space. In an adaptive optics system, a wavefront sensor is used to detect wavefront errors, which are corrected by controlling an adaptive or deformable mirror to counteract them. A deformable mirror can also be used to inject known perturbations into the system in order to probe the wavefront for common-path wavefront sensing [18].

MEMS DMs are one proposed technology to enable the precise wavefront control required for space-based high contrast imaging. There are several other types of DMs that have been used for astronomical applications, which are summarized in Madec 2012 [19], a review of deformable mirror technologies for ground-based astronomy applications. Stacked array PMN and PZT, bimorph, and voice coil actuator technologies are summarized here at a high level. Stacked array DMs use ferroelectric actuators made up of stacks of individual disks. Lead zirconate titanate (PZT) actuators are made up of piezoelectric ceramic disks that elongate with a linear response to the applied voltage [19]. These actuators can provide ∼5 μm of stroke, but exhibit hysteresis effects which makes high-speed, precise control of the actuators difficult [20]. Another type of stacked array DM is based on lead magnesium niobate (PMN) piezolectric ceramics. PMN actuators elongate with a quadratic response to applied voltage, and are sensitive to temperature [19,21]. Adaptive Optics Associates (AOA) Xinetics PMN DMs are currently baselined for the Wide-Field Infrared Survey Telescope (WFIRST) mission and have been used at the NASA Jet Propulsion Laboratory High Contrast Imaging Testbed [22].

Bimorph DMs use the transverse piezoelectric effect to control the curvature of the DM [19]. These DMs offer high reliability, large stroke, and high accuracy [23], but require high voltages and are limited to actuator counts of ∼200–300 [19]. Voice coil actuator DMs use the Lorentz force to control the shape of a thin floating optical shell [19]. Voice coil actuators are typically larger, high power systems to achieve large strokes, and have been used as deformable secondary mirrors for large ground-based telescopes [24].

### 2.3. MEMS Deformable Mirrors

MEMS Deformable Mirrors were first developed in the 1990s [25,26,27]. They are manufactured in batches out of layers of conducting and insulating polysilicon films and then selectively etched to form the actuators which are individually addressable through wire channels on a ceramic carrier [28]. The continuous or segmented mirror surface is coated for optical quality. Each actuator is controlled by applying a voltage to the actuator electrodes which electrostatically attracts the electrically grounded actuator membrane. Figure 1 shows an example of a MEMS DM from Boston Micromachines Corporation (BMC).

MEMS DMs are small in size and have low power requirements which makes them promising for aerospace applications [20]. MEMS DMs can enable extremely high actuator densities with actuator pitches of ∼300–400 μm [29,30] compared to actuator spacings of ∼5–7 mm for PMN DMs [21], which allows for high resolution control with smaller optics throughout the system which can reduce cost [29]. MEMS DMs are manufactured in a bulk machining process at a cost of ∼$100 per actuator, which is an order of magnitude less than the ∼$1000 per actuator cost of conventional DMs [19,31].

MEMS DMs have been shown to be capable of providing subnanometer scale flattening needed for high contrast imaging applications [32,33]. A stability test showed a median of 0.08 nm actuator motion over 40 minutes of operation, and a repeatability test showed that the displacements were repeatable to less than 1 nm surface precision [1]. Laboratory tests have demonstrated their use for phase correction to 6 nm root-mean-square (RMS) residual errors within the controllable spatial frequencies [34]. DMs can also be used to remove phase differences between the arms of an interferometer for speckle nulling applications [35], and a MEMS DM has been demonstrated to reach path-length difference control down to 2 nm RMS [36]. In addition to astronomy, MEMS DMs are used for applications such as biological imaging [37], laser communications [38], and Earth observation imaging [39].

Other types of optical MEMS devices have flown in/near space before, such as the single micromirror used on the MEMS Telescope for Extreme Lightning (MTEL) [40] and the microshutter array used in the Far-UV Off Rowland-circle Telescope for Imaging and Spectroscopy (FORTIS) sounding rocket instrument [41,42]. MEMS Fast Steering Mirrors (FSMs) have been proposed for pointing control in free space laser communications systems [43], and MEMS Digital Micromirror Devices (DMDs) have been tested for space applications to be used as a programmable slit mask for multi-object spectroscopy [44].

#### Suppliers of MEMS DMs

A short summary of a few of the manufacturers of MEMS DMs is included here. Boston Micromachines Corporation (BMC) is a US company that was founded in 1999 as a spin-off from Boston University. BMC MEMS DMs are manufactured as either continuous membrane mirrors or segmented mirrors with actuator counts from 32 up to 4092 actuators [30,45,46]. The BMC MEMS fabrication process uses surface micromachining processes to create an array of independently addressable electrostatic actuators on a silicon substrate as described in [47]. A picture of a BMC MEMS DM is shown in Figure 1. BMC mirrors have been used or are planned for several current and future space projects such as PICTURE, PICTURE-B, PICTURE-C, and DeMi as described in the technology demonstration section of this paper.

IrisAO is a former US company that started in 2002 and supplied segmented MEMS DMs with between 37 and 313 individual actuators with tip/tilt/piston control. IrisAO MEMS DMs were manufactured using polysilicon surface micromachining to create actuator platforms and flip-chip bonding bulk-micromachined mirror segments to the actuator platform arrays. Each actuator has three parallel-plate electrodes to enable piston, tip, and tilt mirror segment motions [2,48,49]. An IrisAO MEMS DM was used for the HiCIBaS high altitude balloon payload as described in the technology demonstration section of this paper.

Flexible Optical B. V. (OKO Tech) is a Dutch company that was founded in 1997 and supplies a variety of adaptive optical devices. They develop Micromachined Membrane Deformable Mirrors (MMDMs) with 17 or 37 actuators. These DMs are bulk micromachined with a flexible membrane stretched over a patterned array of electrodes [50].

Microscale is a US company that started in 2006 focusing on the development of deformable mirrors for NASA’s space-based applications [51,52,53,54]. Microscale products include a PZT stack actuator DM, a PMN-PT stack actuator DM, and a multiplexer-based driver application-specific integrated circuit (ASIC) which would replace bulk DM electronics. The PMN-PT stack actuators are manufactured through a patent-pending bulk-micromachining process in order to achieve better electro-mechanical performance and improved yield. The actuator arrays are directly integrated with a MEMS micromachined SOI (Silicon-on-Insulator) mirror. The multiplexer driver ASIC is designed with ultralow static power dissipation such that an entire Switch-Mode (SM) ASIC consumes 2 mW static power. The Microscale Driver ASIC has 32 × 32 outputs of 40 V capability, and can be extended to larger formats by mosaicking multiple packages. A 64 × 64 driver ASIC module has been demonstrated by mosaicking 4 of the 32 × 32 ASIC packages. The driver ASIC is currently undergoing radiation testing.

Many other designs of MEMS DMs have been researched, including a design with hexagonal parallel plate actuators [55], and a large-stroke design that uses a polyimide thin film membrane for the mirror [56]. Another design uses an electrostatic lever design to create high-stroke tip/tilt/piston actuators and uses a heat treatment process to correct the mirror shape after fabrication [57]. A hybrid design uses MEMS fabrication processes with a PZT ceramic material to create a bulk PZT actuated MEMS DM [58,59].

### 2.4. Ground-Based Astronomy Applications

MEMS DMs have been used successfully for ground-based astronomy applications. The Gemini Planet Imager uses a 64 × 64 actuator BMC MEMS DM [60] and achieved an order-of-magnitude improvement in contrast compared to previous adaptive optics instruments [4]. The GPI mirror underwent extensive qualification and testing before on-sky operations [61].

BMC MEMS DMs have also been deployed in the Robo-AO system at Palomar Observatory (later moved to Kitt Peak) [62], the Shane-AO system at Lick Observatory [63], and the SCExAO system at the Subaru Telescope [64,65]. The upgraded MagAO-X instrument plans to use a 2040-actuator BMC MEMS DM, and a laboratory DM characterization pipeline has been developed to test these mirrors at the University of Arizona Wavefront Control testbed [66].

Ground-based high contrast imaging instruments are likely limited to contrasts of between $10^{-8}$ and $10^{-9}$ due to the effects of Earth’s atmosphere [67,68,69] unless new technology such as bright satellite-based laser guide stars can be used for wavefront sensing [70]. Space-based observatories are the platform of choice to reach the $10^{-10}$ to 10${}^{-11}$ contrasts required to image Earth-like exoplanets around Sunlike stars [12,71]. Proposed starshade missions [72] and satellite laser guidestar missions [70] trade significantly relaxed telescope stability requirements for a new set of manufacturing and formation-flying challenges and are likely best-suited to deep characterization of a few high-priority systems because the maneuvering time for the starshade or laser guidstar satellite to traverse between targets leads to a low observational cadence.

### 2.5. Design Considerations

The goal of an adaptive optics system for space telescopes is to counteract optical aberrations due to thermal and mechanical variability in the telescope system. Mitigating the wavefront errors resulting from these dynamic effects in space typically requires control loops at a few Hz [73], much slower than the 2500 Hz required to compensate for atmospheric aberrations in a ground-based astronomical observatory.

A derivation of how deformable mirror parameters affect wavefront correction performance can be found in [9]. This section summarizes the key results from that chapter, which provide a relevant framework for the missions discussed in more detail in this paper.

The number of actuators across the DM governs the spatial frequency of errors that can be corrected [9]. Since they are able to fit and subtract higher spatial frequency surface errors, high actuator density DMs can correct contrast over a larger angular area with smaller optics along the optical train, which can reduce overall cost [29]. The actual shape and size of the dark hole, or area of high contrast in the science image, depends on the design of the coronagraph system.

The stroke required for the DM is related to the expected amplitudes of the errors the DM needs to correct in the system [19]. Achieving high strokes on MEMS DMs can be a challenge due to the tight actuator spacing, but this can be resolved by using a high stroke, low actuator count non-MEMS DM to correct larger low frequency errors in addition to the lower stroke, high spatial frequency MEMS DM to correct residual high frequency errors in a “woofer-tweeter” configuration [74,75].

Speckles due to phase errors alone can be corrected by a single DM across the image, but speckles due to phase and amplitude errors can only be canceled on one side of the star or the other unless a second DM is used [9]. For high contrast space-based applications, the use of two DMs is planned to enable dark holes on both sides of the star [76]. Several proposed approaches to control the two DMs and optimize contrast across a large area at multiple wavelengths simultaneously are introduced in [77]. Electric Field Conjugation estimates the phase in the pupil and sets the DM to conjugate this phase to flatten the wavefront [78]. Stroke minimization builds up an interaction matrix to describe the mapping between the DM and the speckle field and calculates the DM shape required to reach a target contrast with the minimum stroke. At each iteration, the interaction matrix is recomputed and the target contrast lowered until the dark hole contrast is achieved with the minimum stroke induced on the DM [76,79]. Stroke minimization has been demonstrated with two DMs (one to correct phase errors and one to correct amplitude errors) for monochromatic light [80] and broadband light for the WFIRST Astrophysically Focused Telescope Assets (AFTA) (now Coronagraphic Instrument (CGI)) instrument [81]. A formalism for approaching multi-wavelength correction problems and an approach for using multiple DMs to create a polychromatic null (a high contrast region across multiple wavelengths) is described in [15].

Other important characteristics of DMs are actuator speed, stability, repeatability, hysteresis, and inter-actuator influence. Actuator speed determines how fast the wavefront control loop can respond to disturbances. Testing with a 32×32 actuator BMC continuous facesheet MEMS DM demonstrated fast actuation with a response time of <0.35 ms [35]. The MEMS DM in the Gemini Planet Imager was able to meet the maximum update rate requirement of 2500 Hz to correct for atmospheric turbulence [4,74].

Actuator stability refers to the ability of the actuators to hold their shape over time, while repeatability refers to the ability of the actuator to return to the same position under the same applied voltage consistently. Laboratory testing with a 1024-actuator continuous facesheet BMC MEMS DM demonstrated flat shape stability to 0.08 nm RMS over 40 minutes of operation [1] and median actuator repeatability to 0.046 nm surface [29]. Stability testing of IrisAO 37 piston/tip/tilt actuator MEMS DMs showed that the open-loop flattening error increased by 5.68 nm rms over 29 months of testing [82].

Hysteresis occurs when a system’s response to an input control signal is not perfectly repeatable, but depends on the prior states of the system. For MEMS DMs, this means that the mirror elements actuate to a different position depending on whether the voltage was ramping up or down to a given value. This effect has been tested in the lab with a 1024 actuator BMC continuous facesheet MEMS DM and shown to be negligible [29].

The inter-actuator influence refers to the passive motion of nearby actuators due to the deflection of a single actuator. This property was measured with a 140 actuator BMC continuous facesheet MEMS DM to be 13.5% for the first directly adjacent actuators and <1% for diagonal actuators and second neighboring actuators [83].

A major step in qualifying a DM for high-contrast imaging is assessing how flat it is under closed loop control. Testing with a 1024-actuator continuous facesheet BMC MEMS DM demonstrated flattening in lab to 0.54 nm RMS [1,84].

## 3. Technology Demonstrations

Applying MEMS technology to space missions introduces several challenges due to the launch and space environment. The launch environment requires spacecraft components to withstand acoustic shocks and vibrational loads during launch. Once the spacecraft is in space, the components must operate in vacuum over varying thermal conditions while withstanding the effects of ionizing radiation. This section summarizes the progress of technology demonstrations aimed at preparing this technology for future space telescope missions.

### 3.1. Ground Testing

A key test for space technology is the ability to operate in vacuum. A 32 × 32 actuator BMC MEMS DM was used in vacuum to create a dark hole with $10^{-7}$ contrast with polychromatic light for the EXoplanetary Circumstellar Environments and Disk Explorer (EXCEDE) mission concept study [85,86]. The Visible Nulling Coronagraph (VNC) testbed used an Iris-AO DM with 169 hexagonal segments to demonstrate a dark hole with $10^{-9}$ contrast in vacuum [87].

A set of IrisAO 37 piston/tip/tilt actuator segmented MEMS DMs underwent thermal testing to assess the mirror response to temperature fluctuations between 21–28 ${}^{\circ }$C. The average change in the surface of the flat DMs due to the temperature change was 0.62–1.42 nm rms/${}^{\circ }$C [82].

Another consideration for potential infrared imaging applications is the ability to operate at cryogenic temperatures in order to reduce thermal noise, which can dominate infrared observations. A cryogenic MEMS DM design was developed by BMC by mounting the mirror chip on a silicon board to reduce thermal stress and oxidizing the substrate to form an insulating layer [88]. A 32-actuator prototype cryogenic DM demonstrated no significant change in performance at 95 K and at room temperature [88], and a 1020-actuator cryogenic DM demonstrated successful operation during three cooling cycles between 5 K and 295 K [89]. A 37-actuator membrane DM by OKO Technologies was tested at cryogenic temperatures and demonstrated similar influence functions with surface deflection reduced by 20% at 78 K compared to room temperature [90].

Radiation can affect MEMS devices when dielectric insulating materials trap charge, impacting the device response to electrostatic actuation [91,92]. The deflection of a single actuator of a BMC MEMS DM was measured under total dose radiation levels of 0–3000 Krad (Si) and demonstrated no significant change during radiation testing [92].

### 3.2. Technology Development For Exoplanet Missions Program

The NASA Exoplanet Exploration Office funds a series of Technology Development for Exoplanet Mission (TDEM) awards to demonstrate technologies for exoplanet missions. BMC has been awarded a TDEM grant in order to demonstrate survivability and performance repeatability of the 952-actuator BMC MEMS continuous DM after exposure to launch-like vibration, shock, and acoustic conditions [93]. The testing will be conducted on a set of 952 actuator continuous facesheet DMs at NASA Goddard Space Flight Center’s Environmental Test and Integration Facility [93]. The mirrors will be split into groups and tested to low, medium, and high vibration, acoustic, and shock levels. The mirrors will be tested before and after testing using a combination of instruments available at BMC, NASA JPL’s Vacuum Surface Gauge instrument, and Princeton High Contrast Imaging Laboratory equipment. This testing will assess any changes in performance by measuring influence functions and displacements of each actuator and measuring the achievable dark hole contrast and stability [93].

### 3.3. The PICTURE Missions

The Planet Imaging Concept Testbed Using a Recoverable Experiment (PICTURE) and Planet Imaging Coronagraphic Technology Using a Reconfigurable Experimental Base (PICTURE-B) sounding rockets and the Planetary Imaging Concept Testbed Using a Recoverable Experiment - Coronagraph (PICTURE-C) high-altitude balloon payload are a series of projects with the goal of demonstrating planet imaging technology in a space and near-space environment. The missions are designed to observe the Epsilon Eridani circumstellar disk environment [94]. The science payloads use MEMS DMs to enable wavefront control and high contrast imaging. For a description of the PICTURE optical payload, see Mendillo et al. 2012 [94]. For a description of path length control in the nulling interferometer with the DM, see Rao et al. 2008 [36]. This section summarizes the design and results of the PICTURE sounding rocket mission (launched October 11), the PICTURE-B sounding rocket mission (launched November 2015), and the PICTURE-C high altitude balloon payloads (launching June 2019 and September 2020).

#### 3.3.1. PICTURE Sounding Rocket Design

The PICTURE (and PICTURE-B) science instrument is a lateral-shearing Mach-Zehnder nulling interferometer, which suppresses the light from the target star through destructive interference while transmitting light through closely spaced bright fringes [36,95]. The payload employs a 32 × 32 actuator BMC MEMS DM for wavefront control in order to minimize the path length difference between the two arms of the interferometer. Thus, the DM does not flatten the overall wavefront or increase the image Strehl, but increases contrast by suppressing transmitted light. The payload also uses a piezoelectric fast-steering mirror for optical stabilization. The wavefront control system measures the wavefront with a calibration interferometer that varies the path length differences between the arms of the interferometer to step through a series of phase offsets and measure the resultant fringes. These fringes are used to build up a phase map of the wavefront in the system, which is used to control the DM to flatten the wavefront and minimize the path difference between the two arms of the interferometer.

#### 3.3.2. PICTURE Testing and Flight Results

Prior to launch, the PICTURE payload underwent vibration testing to NASA Vehicle Level 2 levels with the DM in place. The payload was shaken in three axes for 10 seconds per axis according to the spectrum specified in [96]. The DM was tested successfully after the vibration test.

The PICTURE sounding rocket flew on 8 October 2011. Unfortunately, ∼70 seconds into flight the primary telemetry transmitter failed. The fine pointing system was demonstrated but none of the other flight demonstration goals can be confirmed due to lack of data [94].

The MEMS DM survived the launch and recovery, but was not measured in the near-space environment [94]. The telescope primary mirror shattered and was not recovered intact but the nuller assembly survived landing and recovery [94]. Some DM actuators were not working after recovery, but the cause of failure cannot be confirmed due to lack of flight data. The DM and cables could have been damaged in assembly, landing, recovery, transport, or disassembly. Since the mirror and cables were permanently bonded together, the entire assembly, including the Kilo-DM, breakout board, and flex cables were replaced prior to reflight. The primary mirror was also replaced before reflight as PICTURE-B.

#### 3.3.3. PICTURE-B Flight Results

PICTURE-B launched with a new Kilo-DM and a new primary mirror in November 2015 [97]. The PICTURE-B flight experienced a payload anomaly, likely due to shifting of the DM mount during flight. During this flight, the instrument did not advance past the wavefront measurement phase.

The data that was collected demonstrates wavefront sensing and shows that the DM flattened, actuating in the space environment, as shown in Figure 2. Since the full payload survived flight and is still assembled, the mirror has not been individually characterized. In qualitative end-to-end payload tests, which subject the payload to the expected vibrational and atmospheric disturbances across the 0.5 m telescope, the MEMS DM has been observed to respond as expected.

#### 3.3.4. PICTURE-C High Altitude Balloon Project

The PICTURE project is continuing with the Planetary Imaging Concept Testbed Using a Recoverable Experiment—Coronagraph (PICTURE-C) instrument high-altitude balloon payload [98]. This instrument uses a Vector Vortex Coronagraph instead of a nulling interferometer. The project has recently switched away from the baselined DM [99] and has selected a BMC 952 actuator MEMS DM driven by a miniaturized DM controller [75] for high-order wavefront control and is scheduled for launch in 2019.

### 3.4. The High-Contrast Imaging Balloon System

The High-Contrast Imaging Balloon System (HiCIBaS) high-altitude balloon payload is designed to test high contrast imaging equipment and algorithms in a near-space environment at a low cost [100]. This technology demonstration payload uses a IrisAO MEMS DM with 37 hexagonal piston-tip-tilt segments for wavefront control with its Coronagraph Wavefront Sensor (CWS) instrument. For a description of the balloon mission objectives and instrument design, refer to Cote et al. 2018 [100]. The full payload flew successfully on 25 August 2018 [101]. During flight, a cabling issue prevented commands from being sent to the DM, so in-flight operation cannot be confirmed. Images taken with a calibration source during flight agreed well with images taken in lab with the DM powered off, which implies that the DM did not move significantly during flight. The DM is still operational, with a single stuck actuator and a small loss of stroke measured post flight. On sky testing of the Low Order Wavefront Sensor (LOWFS) was conducted in 2018 to validate the wavefront sensors and compare the measurements from external starlight and the internal laser source [102]. These tests also compared the signals from the payload Shack-Hartmann wavefront sensor and the pyramid wavefront sensor and showed good agreement between all configurations [102].

### 3.5. The Deformable Mirror (DeMi) CubeSat Payload

The Deformable Mirror Demonstration Mission (DeMi) CubeSat payload will demonstrate the on-orbit performance of a 140-actuator BMC MEMS DM on a 6U (10 cm × 20 cm × 30 cm) CubeSat [103,104,105,106]. The goal of this mission is to raise the Technology Readiness Level (TRL) of MEMS DM technology from a TRL of 5 to at least a TRL of 7 for future space telescope applications [107]. For a description of the DeMi optical design, see Allan et al. 2018 [108]. For a description of the DM driver development, see Haughwout 2018 [20]. The key DeMi mission requirements are to measure individual DM actuator wavefront displacement contributions to a precision of 12 nm, measure low order optical aberrations to λ/10 accuracy and λ/50 precision, and correct static and dynamic wavefront errors to less than 100 nm RMS error. This section summarizes the design and concept of operations for the mission as well as integration and testing progress in preparation for the expected mission launch in late 2019.

#### 3.5.1. Design

The DeMi optical design contains an off axis parabola-based telescope with a 50 mm primary mirror, a 140-actuator BMC Multi DM, and both an Image Plane wavefront sensor (WFS) and a Shack Hartmann WFS [108]. The Image Plane WFS captures pictures of the system Point-Spread Function (PSF) and serves as a truth sensor for wavefront correction. The Image Plane WFS is also used to detect tip-tilt errors. The Shack Hartmann WFS uses a lenslet array to split the light into sections and focus it into a grid of spots on the detector. The displacement of each spot corresponds to the wavefront slope incident on the corresponding lenslet. This sensor is used to measure wavefront aberrations and monitor the DM health on-orbit. A diagram of the DeMi optical payload is shown in Figure 3. The original driver electronics for the DM were not designed for space operation and were too large to fit onboard the 6U CubeSat, so custom miniaturized drive electronics based on commercial components were developed [20].

The DeMi payload has both external and internal operational modes. It can observe stellar targets and collect photometric measurements through the primary aperture, or it can use the internal laser fiber source for calibration measurements. For external observations of stellar targets, the Image Plane WFS will be used for closed-loop control of the DM, and performance will be measured with the Shack Hartmann WFS. For internal calibration, the internal laser source will be turned on and either the Image Plane WFS or the Shack Hartmann WFS will be used for closed-loop control of the DM, with the other sensor measuring performance. The internal laser source will also be used for actuator tests of the DM where each actuator will be poked and the resultant wavefront will be measured on both the Image Plane WFS and the Shack Hartmann WFS. This data will be used to assess the on-orbit performance of the MEMS DM over a year of operation.

#### 3.5.2. Integration and Testing

The DeMi payload is currently in the integration and testing stage at MIT. A full Engineering Model (EM) has been assembled and aligned for end-to-end system testing. This was used to check the integration procedures and find any mechanical issues before the flight unit is built [109]. Figure 4 shows the aligned engineering model of the DeMi optics payload. The miniaturized DM driver electronics are shown in Figure 5. Figure 6 shows the result of a fit check with the DeMi optics bench, DeMi electronics, and Blue Canyon Technologies bus simulator within the 6U spacecraft chassis from Blue Canyon Technologies.

The Engineering Model has been used to test the voltage response of the DM actuators with the wavefront sensors [107]. For this test, a single actuator or a 4 × 4 region was displaced to a certain voltage and measured on the Shack Hartmann WFS. The spot displacements on the Shack Hartmann WFS were used to measure the actuator displacement which were compared to values reported from BMC testing. Results from this test are shown in Figure 7. The mean difference between the EM measurements and the BMC values was 28.9% for single actuator pokes and 19.4% for 4 × 4 regions.

The engineering model is also being used to test the miniaturized COTS-based DM driver developed for DeMi. Figure 8 shows the results of displacing a single actuator with the DeMi driver compared to displacement measurements from the BMC driver. The mean difference between the DeMi driver and BMC driver displacements was 11.7%.

The next steps for the DeMi project are full environmental testing, wavefront sensing and correction tests, flight model integration and test, and integration with the spacecraft bus in preparation for launch. Launch is expected in late 2019. The flight results from this mission will demonstrate MEMS DM technology in space so it can be applied to future space telescope high contrast imaging applications, which are summarized in the next section.

## 4. Technology Development and Path Forward

Deformable mirrors are a key technology for future space telescopes using coronagraphic instruments for high-contrast imaging of exoplanets.

Several sounding rocket, space balloon, and CubeSat missions have flown or are flying in the near future to prepare this technology for space mission applications. The PICTURE and PICTURE-B sounding rocket missions have flown MEMS DMs and demonstrated their operation and actuation in the near space environment. The PICTURE-C high altitude balloon mission aims to demonstrate $10^{-7}$ contrast using a vector vortex coronagraph, image plane wavefront sensor, and a 952-actuator BMC MEMS DM. The HiCIBaS balloon mission hopes to demonstrate a 37-segment IrisAO MEMS DM with hexagonal piston-tip-tilt segments at high altitudes. The HiCIBaS balloon mission and DeMi CubeSat mission aim to raise the TRL for MEMS DM technology from 5 to 7. The DeMi mission aims to demonstrate wavefront control in space with a 140-actuator BMC MEMS DM.

There are several outstanding challenges in applying this technology to space missions. One of the key challenges is developing DM driver electronics that meet the size, weight, and power constraints of space missions and can withstand the radiation environment of a long duration space flight mission. The DeMi team and a team at NASA Ames Research Center have designs for DM driver electronics that meet these requirements [20,75]. Another challenge is handling tip-tilt errors alongside higher order optical aberrations with the DM. A proposed solution to this problem is to develop DM mounts with tip-tilt control so these aberrations can be corrected without adding extra path length to the optical design.

MEMS DMs have been proposed or baselined for several future space missions. The NASA Small Business Innovation Research Program has funded the development of high actuator count MEMS DM devices at BMC for future exoplanet technology applications. Last year, BMC announced the production of a fully-functioning 2040-element MEMS DM through this program which is being evaluated for potential inclusion on the WFIRST CGI instrument [110]. Two of the mission concept studies for the upcoming Astrophysics Decadal Survey have exoplanet direct imaging instruments that use DMs for wavefront control. The Habitable Exoplanet (HabEx) mission study baselines two 64 × 64 MEMS DMs for wavefront control [111], while the Large UV-Optical-InfraRed Surveyor (LUVOIR) mission study baselines two high-actuator count BMC MEMS DMs and identifies these devices as a key technology area for the mission [112,113,114].

Looking forward, MEMS DMs show potential to enable high-contrast imaging of exoplanets for future space telescopes. This paper summarizes the status of technology demonstrations to advance this technology for space telescope applications.

## Figures and Tables

**Figure 1 micromachines-10-00366-f001:**
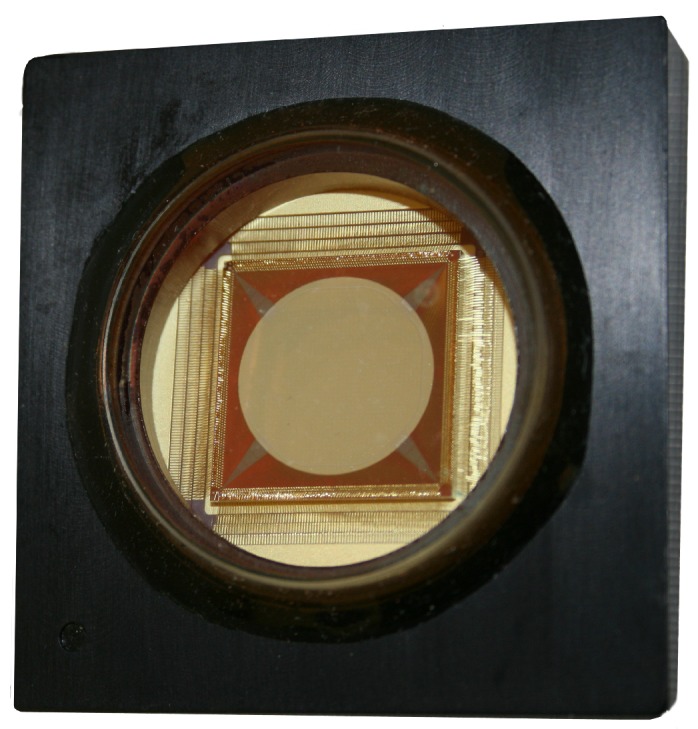
MEMS Deformable Mirror from Boston Micromachines Corporation. The mirror pictured is a KiloDM with 952 actuators across the 1 cm-wide aperture.

**Figure 2 micromachines-10-00366-f002:**
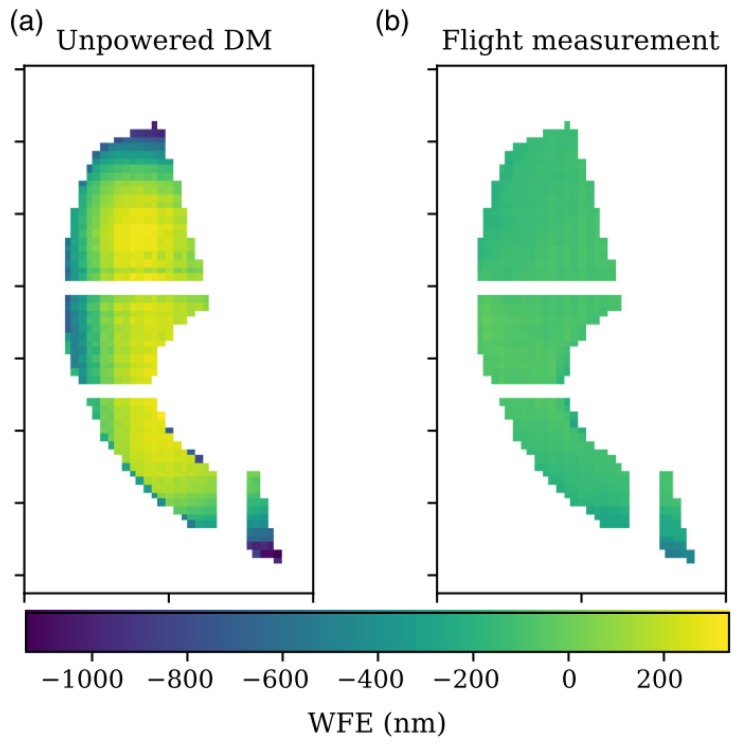
PICTURE-B flight measurement of the deformable mirror surface (**b**) compared to the unpowered DM profile, for the same unit, provided by BMC (**a**) demonstrates that the BMC Kilo MEMS DM was turned on and operating in the near-space environment. The median system stability measured in flight, including telescope and coronagraph wavefront errors other than tip, tilt, and piston, was 3.6 nm per pixel. Data is masked under the shearing coronagraph’s Lyot mask and for a region where wavefront sensor visibility was low. Reproduced from Douglas et al. [97].

**Figure 3 micromachines-10-00366-f003:**
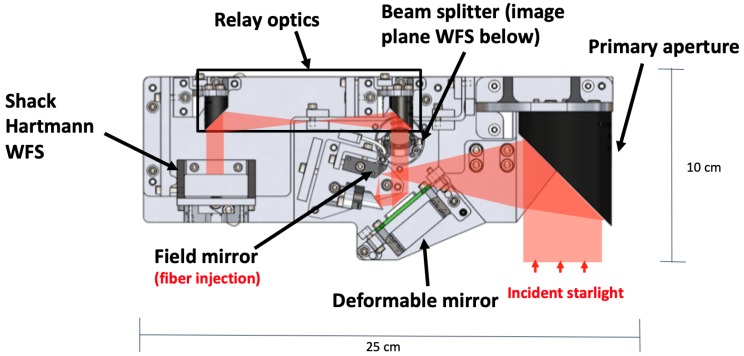
Diagram of the optical components for the Deformable Mirror Demonstration Mission (DeMi) CubeSat payload with ray path overlaid.

**Figure 4 micromachines-10-00366-f004:**
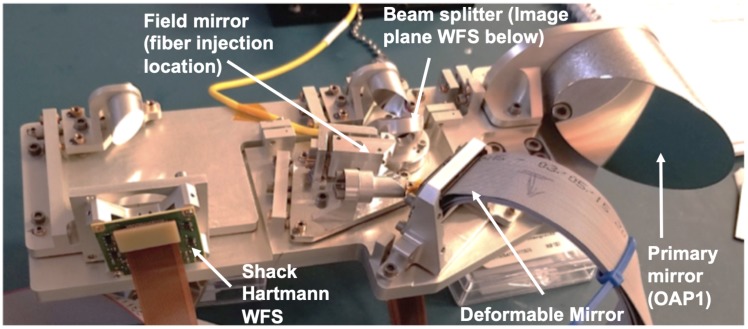
Engineering Model for the Deformable Mirror Demonstration Mission (DeMi) CubeSat Payload.

**Figure 5 micromachines-10-00366-f005:**
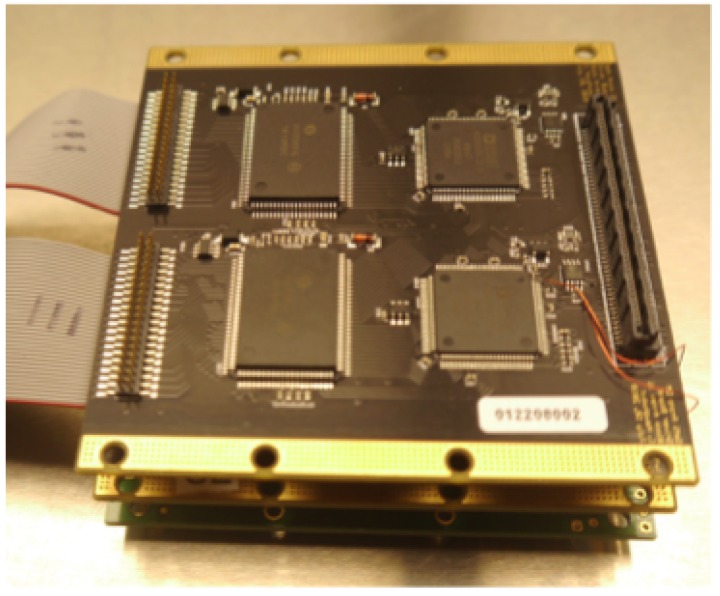
Miniaturized DM driver electronics for DeMi CubeSat payload as described in [20]. The dimensions of each board are 80 mm × 80 mm × 10 mm. The complete driver stack consists of two driver boards, the high voltage power supply board, and the two Raspberry Pi boards. Each driver board has 70 output channels with 14 bits of resolution from 0 to 180 Volts.

**Figure 6 micromachines-10-00366-f006:**
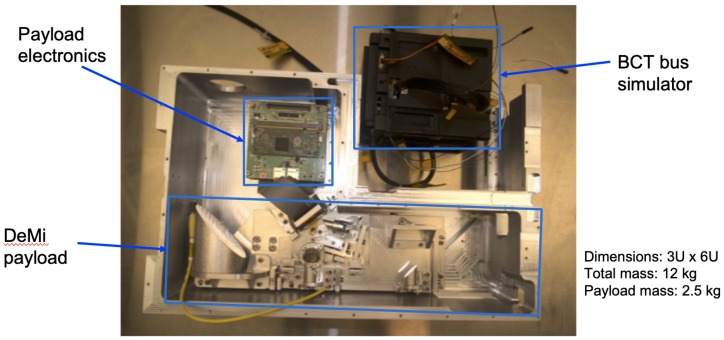
DeMi optical bench and electronics fit check in 6U bus chassis from Blue Canyon Technologies. Entrance baffle (top left), camera controller boards (bottom right) and flight bus stack are omitted.

**Figure 7 micromachines-10-00366-f007:**
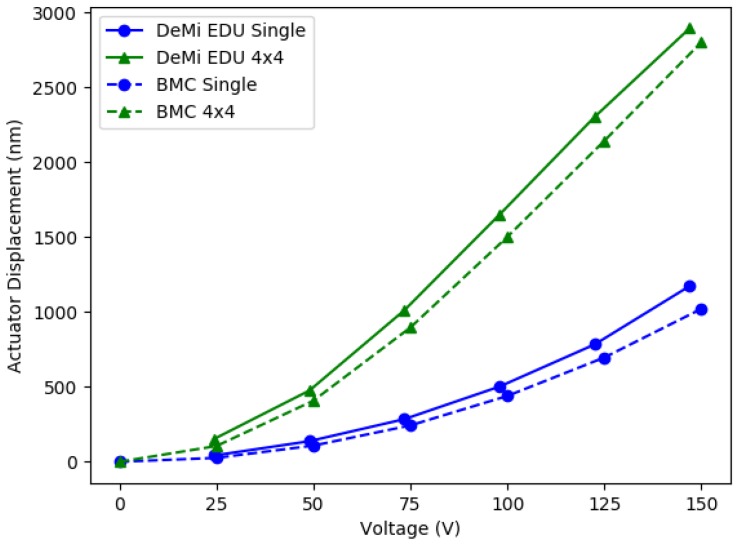
Results from actuator displacement tests. Plot shows actuator displacements measured from spot displacements on the Shack Hartmann WFS vs. the input voltage. The solid lines show measurements from the engineering model and the dashed lines are reported values from BMC. The mean difference between the EM measurements and the BMC values was 28.9% for single actuator pokes and 19.4% for 4 × 4 regions.

**Figure 8 micromachines-10-00366-f008:**
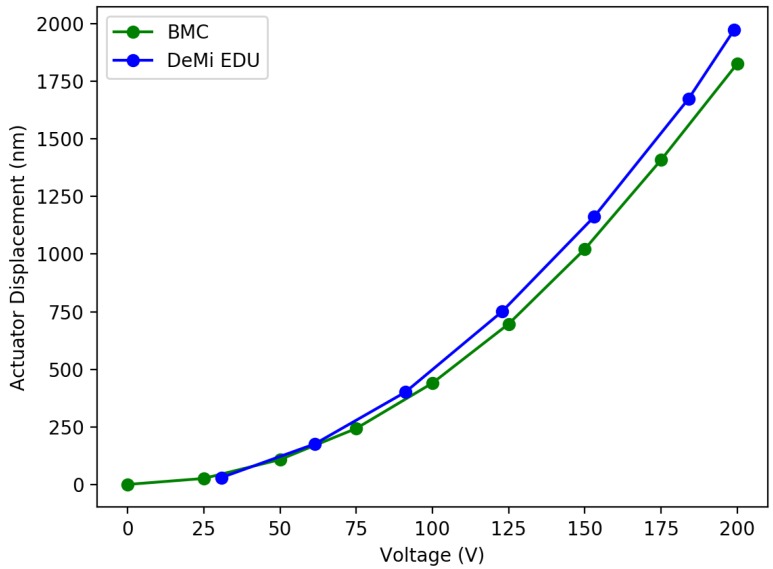
Results from miniaturized DM driver tests. Plot shows single actuator displacements measured from spot displacements on the Shack Hartmann WFS vs. the input voltage. Blue line is DeMi COTS-based miniaturized driver performance and green line is reported values from BMC. The mean difference between the DeMi driver and BMC measurements was 11.7%.

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
