# Peer review of "MEMS Deformable Mirrors for Space-Based High-Contrast Imaging"

_micromachines, 2019, doi:10.3390/mi10060366_

Round 1

Reviewer 1 Report

The paper by Morgan et al presents short overview of DM technologies, a very clear review of the state of space readiness testing of MEMS DMs, in particular BMC devices. 

The manuscript is very well written, and I have only a few minor comments on typos and requests for clarification, detailed below. 

My main concerns with the paper are 

a) that I am not 100% sure how much the manuscript is meant as a comprehensive review of available technologies, versus on outline of the ongoing efforts to raise the TRLs of MEMS DMs, or description of results from these tests. The latter are very sparse, due to the challenging nature of those experiments. Not much is learned from the failure modes of the experiments that are described at this point, as they don't pertain to the actual MEMS mirror. Maybe a short summary of the outcomes would be useful, to highlight the knowledge gained in those tests. 

b) that the capabilities of, and the use of technolgy from, BMC are described in some detail, above what is directly related to the described experiments. While this by itself is no problem, of course, and BMC has a rather unique position in the field, the fact that te CEO is a co-author makes me wonder if it would be better to include more alternatives, or remove some of the specific examples, to avoid possible concerns by readers that the field is not presented in a balanced way. I am not trying to say this is indeed the fact, but think it is an-often voiced concern that would take away from the paper. 

Detailed comments (always starting with line number):

8. PICTURE used twice. 

33. "Process" 2x

57. "flatness of indivi. atoms" - this part of the sentence seems out of context.

102. "influence" should be explained

97-108: reconcile numbers, or explain differences: 6nm RMS, 2nm OPD, 1nm...

123. "between" 2x

125. "Lighning" typo

134. "Traub+Opp..., in Sarah Seager's book Exoplanets.." I find that a very colloquial way to cite this source, could be changed for style. 

140. Dark hole: define

143. The woofer-tweeter config does not solve the challenge to get high stroke on a DM, per se. Maybe another sentence to clarify how that works (high stroke only needed over larger spatial scales) would clarify things.

160. polychr. null: define

173. in lab > in the lab

Fig 1: why is the pupil shape so odd? A short explanation of the shape would be helpful.

276. The terminology of image plane WFS is new to me. I unerstand this detector to be a simple imager that records the shape of the PSF. I've heard the phrase truth sensor before. Consider explaining this a bit better, and include a few words on what technique is used to derive the wavefront shape from measuring the PSF?

Also: consider not using tip-tilt error as example for aberration, as in AO it is often trated separately

Fig 2: drawing the beam path would be very useful!

298. "and aligned engineering model": there seems to be a word missing

325. "TLR 6 to 7": earlier in the manuscript. TLR 5 up to 7 is mentioned?

328. are > is

332. "low-order tip tilt" > "tip tilt" (there is no high order tip tilt..)

para. 335 ff: unclear to me what this paragraph is exactly meant to convey.

Author Response

p.p1 {margin: 0.0px 0.0px 0.0px 0.0px; font: 12.0px 'Helvetica Neue'} p.p2 {margin: 0.0px 0.0px 0.0px 0.0px; font: 12.0px 'Helvetica Neue'; min-height: 14.0px}

Dear reviewer, 

Thank you very much for your detailed and helpful comments! I have addressed your edits and suggestions and I think they made the paper much better. Thanks for your thoughtful suggestions! The changes I made addressing your comments are summarized below. 

Best,
Rachel Morgan 

a) that I am not 100% sure how much the manuscript is meant as a comprehensive review of available technologies, versus on outline of the ongoing efforts to raise the TRLs of MEMS DMs, or description of results from these tests. The latter are very sparse, due to the challenging nature of those experiments. Not much is learned from the failure modes of the experiments that are described at this point, as they don't pertain to the actual MEMS mirror. Maybe a short summary of the outcomes would be useful, to highlight the knowledge gained in those tests. 

I adjusted the wording of the results to emphasize the knowledge gained from the previous technology demonstrations. I also expanded the discussion of ground environmental testing to provide more details on expected performance. 

b) that the capabilities of, and the use of technolgy from, BMC are described in some detail, above what is directly related to the described experiments. While this by itself is no problem, of course, and BMC has a rather unique position in the field, the fact that te CEO is a co-author makes me wonder if it would be better to include more alternatives, or remove some of the specific examples, to avoid possible concerns by readers that the field is not presented in a balanced way. I am not trying to say this is indeed the fact, but think it is an-often voiced concern that would take away from the paper. 

I expanded my discussion of other technologies/vendors to make the paper more balanced. I also added another co-author from Microscale, another vendor of DMs that is developing systems for space applications.

Detailed comments (always starting with line number):

8. PICTURE used twice. 

Fixed typo. 

33. "Process" 2x

Fixed typo. 

57. "flatness of indivi. atoms" - this part of the sentence seems out of context.

Fixed typo. 

102. "influence" should be explained

Added sentence to clarify

97-108: reconcile numbers, or explain differences: 6nm RMS, 2nm OPD, 1nm...

Moved this section and reorganized to make the different tests more clear 

123. "between" 2x

Fixed typo. 

125. "Lighning" typo

Fixed typo. 

134. "Traub+Opp..., in Sarah Seager's book Exoplanets.." I find that a very colloquial way to cite this source, could be changed for style. 

Changed the phrasing of the citation

140. Dark hole: define

Defined in the sentence

143. The woofer-tweeter config does not solve the challenge to get high stroke on a DM, per se. Maybe another sentence to clarify how that works (high stroke only needed over larger spatial scales) would clarify things.

Expanded this description to clarify

160. polychr. null: define

Defined 

173. in lab > in the lab

Fixed typo. 

Fig 1: why is the pupil shape so odd? A short explanation of the shape would be helpful.

Added description to figure caption

276. The terminology of image plane WFS is new to me. I unerstand this detector to be a simple imager that records the shape of the PSF. I've heard the phrase truth sensor before. Consider explaining this a bit better, and include a few words on what technique is used to derive the wavefront shape from measuring the PSF?

Also: consider not using tip-tilt error as example for aberration, as in AO it is often trated separately

Defined image plane WFS and how it is used on the system. It is a specific component of our spacecraft so our use of the phrase may vary from other applications but the definition given explains what we mean by “image plane sensor” 

Fig 2: drawing the beam path would be very useful!

Drew in on figure

298. "and aligned engineering model": there seems to be a word missing

Fixed typo. 

325. "TLR 6 to 7": earlier in the manuscript. TLR 5 up to 7 is mentioned?

Reconciled difference between mission documentation

328. are > is

Fixed typo. 

332. "low-order tip tilt" > "tip tilt" (there is no high order tip tilt..)

Fixed typo. 

para. 335 ff: unclear to me what this paragraph is exactly meant to convey.

I expanded this paragraph and tried to organize conclusion better to make it more clear. 

Reviewer 2 Report

This manuscript can be accepted as  is.

Author Response

p.p1 {margin: 0.0px 0.0px 0.0px 0.0px; font: 12.0px 'Helvetica Neue'} p.p2 {margin: 0.0px 0.0px 0.0px 0.0px; font: 12.0px 'Helvetica Neue'; min-height: 14.0px}

Dear reviewer, 

Thank you very much for your time reviewing the paper! I have made a few changes to the document to emphasize the knowledge gained from ground testing and space demonstrations of MEMS DMs and describe the different options of MEMS DMs in more detail. 

Thanks,
Rachel Morgan